



# FOREWARNS: Development and multifaceted verification of enhanced regional-scale surface water flood forecasts

Ben Maybee[1], Cathryn E. Birch[1], Steven J. Böing[1], Thomas Willis[2], Linda Speight[3], Aurore N. Porson[4], Charlie Pilling[5], Kay L. Shelton[6] and Mark A. Trigg[7]

[1]School of Earth and Environment, University of Leeds, Leeds, LS2 9JT
[2]School of Geography, University of Leeds, Leeds, LS2 9JT
[3]School of Geography and the Environment, University of Oxford, Oxford, OX1 3QY
[4]Met Office, Exeter, EX1 3PB
[5]Flood Forecasting Centre, Exeter, EX1 3PB
[6]JBA Consulting, Skipton, BD23 3FD
[7]School of Civil Engineering, University of Leeds, Leeds, LS2 9JT

*Correspondence to*: Ben Maybee, b.w.maybee@leeds.ac.uk

**Abstract.** Surface water flooding (SWF) is a severe hazard associated with extreme convective rainfall, whose spatial and
temporal sparsity belies the significant impacts it has on populations and infrastructure. Forecasting the intense convective rainfall that causes most SWF on the temporal and spatial scales required for effective flood forecasting remains extremely challenging. National scale flood forecasts are currently issued for the UK and are well regarded amongst flood responders, but there is a need for complimentary enhanced regional information. Here we present a novel SWF forecasting method, FOREWARNS (*Flood f**ORE**casts for Surface **WA**ter at a **R**egio**N**al **S**cale*), that aims to fill this gap in forecast provision.
FOREWARNS compares reasonable worst-case rainfall from a neighbourhood-processed, convection-permitting ensemble forecast system against pre-simulated flood scenarios, issuing a categorical forecast of SWF severity. We report findings from a workshop structured around three historical flood events in Northern England, in which forecast users indicated they found the forecasts helpful and would use FOREWARNS to complement national guidance for action planning in advance of anticipated events. We also present results from objective verification of forecasts for 82 recorded flood events in Northern
England from 2013–2022, and for 725 daily forecasts spanning 2019–2022, using a combination of flood records and precipitation proxies. We demonstrate that FOREWARNS offers good skill in forecasting SWF risk, with high spatial hit rates and low temporal false alarm rates, confirming that user confidence is justified, and that FOREWARNS would be suitable for meeting the user requirements of an enhanced operational forecast.

*Summary: This paper presents the development and verification of FOREWARNS, a novel method for regional-scale*
*forecasting of surface water flooding. We detail outcomes from a workshop held with UK forecast users, who indicated they valued the forecasts and would use them to complement national guidance. We use results of objective forecast tests against flood observations over Northern England to show that this confidence is justified, and that FOREWARNS meets the needs of UK flood responders. (489/500 characters)*



# 1 Introduction

Surface water flooding (SWF) represents all pluvial flooding caused directly by intense rainfall, prior to water entering natural or man-made drainage networks or watercourses (Speight et al., 2021). In the United Kingdom (UK) such events are treated distinctly from in-channel fluvial flash floods, but both hazards pose similar challenges to forecasters and responders. More UK properties are at risk from SWF than from the combined risk from river and sea (DEFRA, 2018), while the frequency and severity of SWF events is expected to increase under current climate projections (Kendon et al., 2014; Chen et al., 2021). In

urban areas with impermeable ground surfaces these events can readily cause major incidents (Green et al., 2017), while the formation of rapid torrents in steep terrain can lead to hazardously high peak-flow rates (Archer and Fowler, 2018). The majority of short-duration, high intensity precipitation in the UK falls from summertime convective weather systems (Hand et al., 2004), which are challenging to forecast accurately at the temporal and spatial resolutions required for flood warnings (Golding et al., 2016).

Ensemble forecast systems quantify this uncertainty, provide a range of possible future outcomes, and allow forecasters to assess the probability of different scenarios (Speight et al., 2021). This is of particular importance for the prediction of extreme events (Swinbank et al., 2016; Hawcroft et al., 2021). Regional convection-permitting ensemble forecasts are now a mature feature of numerical weather prediction (NWP) systems at meteorological services globally (Brousseau et al., 2011, 2016; Beck et al., 2016; Klasa et al., 2018; Frogner et al., 2019a, b), and they have increasingly been adopted as the drivers for

hydrological models and SWF forecasting (Golding et al., 2016; Speight et al., 2018, 2021; Wu et al., 2020).

The level of complexity involved in a SWF forecast can vary from using simple empirical triggers to real-time hydraulic modelling (Henonin et al., 2013; Speight et al., 2021). High resolution modelling is appropriate on nowcasting (below 6 hours) time scales (Yu and Coulthard, 2015; Coles et al., 2017; Green et al., 2017), but at longer lead times high uncertainties in the driving rainfall limit the utility of such forecasts (Moore et al., 2015; Flack et al., 2019; Birch et al., 2021). A suitable

compromise is to link rainfall forecasts to pre-simulated impact scenarios, built from offline modelling, by using look-ups. This method has been successfully applied at urban, regional and national scales (Dottori et al., 2017; Saint-Martin et al., 2016; Speight et al., 2018), including in operational settings (Pilling, 2016; Aldridge et al., 2020).

Scientific and technical feasibility of a SWF forecasting method are necessary, but not sufficient, criteria for operational adoption: the requirements of end-users must also be balanced (Flack et al., 2019). The flood response community has a

growing familiarity with, and appetite for, probabilistic forecasts which convey uncertainty (Demeritt et al., 2013). The growth of impact-based forecasting enables good decision making despite high uncertainties, focussing attention on areas with highest expected impacts (Ramos et al., 2013; Merz et al., 2020). Many studies have investigated the added-value for flood responders of new SWF forecasting tools, typically by using individual interviews, group workshops and discussion, and interactive exercises (Frick and Hegg, 2011; McEwen et al., 2014; Arnal et al., 2016, 2020; Ochoa-Rodríguez et al., 2018). Including

users in the co-design of new products, combined with communication and training, is an essential step of forecast development (Golding, 2022).



A further step required to enable users to optimise decision making, and to improve objective skill, is forecast verification. Verification is routinely conducted for operational ensemble forecasts of precipitation, typically against continuous rain gauge and radar observations (Raynaud et al., 2019; Porson et al., 2020). For hazardous extremes such as SWF, however, verification
methods are often lacking due to the sparsity of reliable and accurate observations against which to test forecasts, and the relative extremity and hence rarity of the events themselves (Welles et al., 2007). For river flooding, in-situ gauges and satellite-derived spatial datasets (e.g. GLoFAS, Alfieri et al., 2013) provide appropriate observations, however these sources typically have limited application to flash flooding and SWF (Gourley et al., 2012; Speight et al., 2021). Records may instead be generated by using rainfall accumulations exceeding predefined thresholds as flood proxies, but such datasets are method
dependent and can differ significantly from records of known floods (Herman and Schumacher, 2018). Promising complimentary datasets are crowdsourced observations, obtained from public reporting systems (Kirk et al., 2021; Macleod et al., 2021), news reports (Archer et al., 2019; Jackson, 2023), or social media (de Bruijn et al., 2018, 2019). The potential of crowdsourced data for supporting verification has been widely acknowledged and demonstrated (Smith et al., 2017; Witherow et al., 2018; Zhang et al., 2019; Macleod et al., 2021). Given the sparsity of alternative observations, such crowdsourced
evidence is critical for supporting the operational adoption of new forecasting systems.

Here we present the development and verification of a novel SWF forecasting method, FOREWARNS (*Flood fOREcasts for surface WAter at a RegioNal Scale*), which neighbourhood-processes output from a convection-permitting ensemble NWP system to compute reasonable worst-case rainfall scenarios (Böing et al., 2020), and compares these scenarios against rainfall drivers of pre-simulated flood modelling. Our aim is to meet the UK user need for enhanced regional guidance that
complements existing national-scale products (Ochoa-Rodríguez et al., 2018), while offering useful guidance beyond existing static SWF risk mapping (Aldridge et al., 2020; Birch et al., 2021). We present findings from a case-study based workshop with forecast users, the outcomes of which informed the development of new objective verification methodologies designed to operate within the constraints of limited observations, rarity of forecast events, and the need to evaluate both spatial and temporal forecast performance. We conduct verification for samples of 82 recorded flood events and 725 daily forecast
issuances, computing categorical measures of forecast performance against flood records and proxies. Combining these results with workshop outcomes, we present a complete picture of the suitability of FOREWARNS for enhanced operational SWF forecasting.

## 2 Methods

### 2.1 Operational forecasting capabilities and responsibilities in the UK

SWF forecasting systems in the UK are tailored towards supporting users who are responsible for incident planning and responding to flood events, and to those issuing subsequent public warnings. Lead Local Authorities (UK local government) are responsible for SWF incident management and therefore expertise can vary widely between regions (Ochoa-Rodríguez et al., 2018). This stands in contrast to fluvial flooding, for which the Environment Agency, a national public body responsible



for protecting and improving the environment, has primary responsibility. Whilst the Environment Agency does not have a statutory responsibility for SWF, they are involved in some SWF response, in coordination with Local Authorities, who are also supported by the Emergency Services (typically Fire and Rescue).

The Met Office National Severe Weather Warning Service (NSWWS) provides public warnings for all severe weather, including thunderstorms and associated impacts (Neal et al., 2014). Warnings are issued for individual hazard types based on ensemble NWP output (Roberts et al., 2023) and expert forecaster judgment. National warnings of SWF are typically only issued when significant impacts are expected, so do not usually cover minor events.

The NSWWS service is informed by the UK regional Met Office Global and Regional Ensemble Prediction System (MOGREPS-UK), a 5 minute temporal resolution, convection-permitting ensemble forecast covering north-western Europe, with a 2.2 km inner domain spanning the UK and Ireland (Hagelin et al., 2017; Porson et al., 2020). The model is nested within a global ensemble, MOGREPS-G, which yields boundary and initial conditions for regional members. These are further centred (Tennant, 2015) on analyses from the 4D-Var data assimilation system of the UKV 1.5 km resolution deterministic model (Milan et al., 2020), while ensemble perturbations are generated using RP2-scheme stochastic physics (McCabe et al., 2016). All FOREWARNS forecasts used in this paper are generated from post-processed MOGREPS-UK ensembles. Flood forecasting is a key application of MOGREPS-UK, and its development has occurred in concert with that of the UK's SWF capabilities (Golding et al., 2016; Hagelin et al., 2017; Speight et al., 2018).

Since 2019, three regional ensemble members have been available in hourly cycles, initialised from the most recent UKV analyses. Typically, members from six cycles are grouped to generate an 18 member time-lagged ensemble which contains a single unperturbed control member and spans a common forecast period of 120 hours (Porson et al., 2020). Prior to 2019 the ensemble comprised 12 members, initialised every 6 hours, and spanning a forecast period of up to 54 hours (Hagelin et al., 2017). The post-2019 system offers the advantages of being updated hourly and having ensembles that incorporate observations from additional high-resolution analyses. Under objective verification, this has led to improved skill scores and ensemble spread (Porson et al., 2020).

MOGREPS-UK rainfall forecasts are routinely used to inform the Flood Guidance Statement (FGS), issued for England and Wales by the Flood Forecasting Centre (FFC), which is the current primary source of flood risk warnings for flood responders. The FGS provides daily Local Authority level guidance for fluvial, pluvial, groundwater and coastal flooding, and is disseminated to all flood responders and some community groups (Pilling, 2016). The existing audience for FGS warnings is taken as the intended user for FOREWARNS.

FGS guidance is issued daily on a four-category scale for each type of flooding, covering lead times of 0 to 4 days. Areas of concern are highlighted by operational forecasters and accompanied by risk matrices and textual detail. The forecast is informed by ensemble outputs from a 1 km resolution Grid-to-Grid (G2G) rainfall-runoff model (Bell et al., 2009; Cole et al., 2015), driven by MOGREPS-UK. Assessments of SWF risk are informed by the Hazard Impact Model (SWFHIM; Aldridge *et al.*, 2020), which processes G2G outputs to select the most appropriate grid-point representative from a set of pre-calculated impact maps from multiple sources. For national FGS guidance these are aggregated to Local Authority level risk areas. SWF





risk at a given location is assessed using UK Risk of Surface Water Flooding (RoSWF) maps, a national dataset of street-level SWF risk (publicly accessible at https://check-long-term-flood-risk.service.gov.uk/map). RoSWF mapping is informed by

modelling of nine flood scenarios (1, 3 and 6 hour accumulations, for 30, 100 and 1000 year return period events), which are aggregated to form compound hazard maps (Environment Agency, 2019).

## 2.2 Enhanced SWF forecasting: FOREWARNS

The FOREWARNS forecasting method is an evolution of Böing et al., 2020, and Birch et al., 2021, and comprises two distinct components: (1) post-processing of MOGREPS-UK rainfall to generate Reasonable Worst-Case Rainfall Scenarios

(RWCRSs) of possible exceptional accumulations; and (2) a look-up comparison of these scenarios against hydrological reference data to quantify the likely severity of SWF associated with the forecast rainfall. The outputs from each scenario's look-up are then combined to generate a single forecast map.

### 2.2.1 Reasonable Worst-Case Rainfall Scenarios (RWCRSs)

To generate RWCRSs we use the neighbourhood post-processing method presented in Böing et al., 2020. For each grid-point,

the method samples a specified threshold percentile $p$ of the maximum accumulations in period $T$, for all neighbouring points within a radius $r$. In this study all accumulation periods start within the same calendar day, but this requirement could be varied. The processing may be conducted either on a single rainfall field, or across multiple ensemble member fields (covering common forecast periods) – any RWCRS is then parametrised as ($r$, $p$, $T$). The timings of maximum accumulation periods may also be extracted and used for forecasting (not featured in this study). As with all neighbourhood processing methods,

output forecast fields are smoothed, which is appropriate for fields with high degrees of spatial uncertainty, such as convective rainfall (Schwartz and Sobash, 2017).

A major benefit of the RWCRS method is that it is applicable to radar observations, allowing verification of post-processed forecasts against comparable observations. We use 5 minute, 1 km grid composite Met Office Nimrod data, a blended radar and nowcasting/NWP product which is verified by rain gauges (Harrison et al., 2000, 2012). At high accumulations, radar-

derived rainfall observations can suffer from substantial quantitative errors (Harrison et al., 2012), but cannot be replaced with gauge data for extreme events due to the limited spatial coverage of gauge networks (Mittermaier, 2008). The possibility of erroneous grid-point extremes in radar (or forecast) fields motivates focussing on high-percentile accumulations rather than maximum values. Figure 1 shows an example ensemble RWCRS forecast and neighbourhood-processed and unprocessed radar observations for Northern England on 30/07/2019. Three distinct bands of intense rainfall occurred (Fig. 1c); these features

caused severe SWF across a large area around the town of Leyburn (Kendon, 2019). The smooth fields output by the neighbourhood processing when applied to the forecast and radar observations may be clearly seen (Figs. 1a,b) relative to the unprocessed observed maximum daily rainfall accumulations (Fig. 1c).


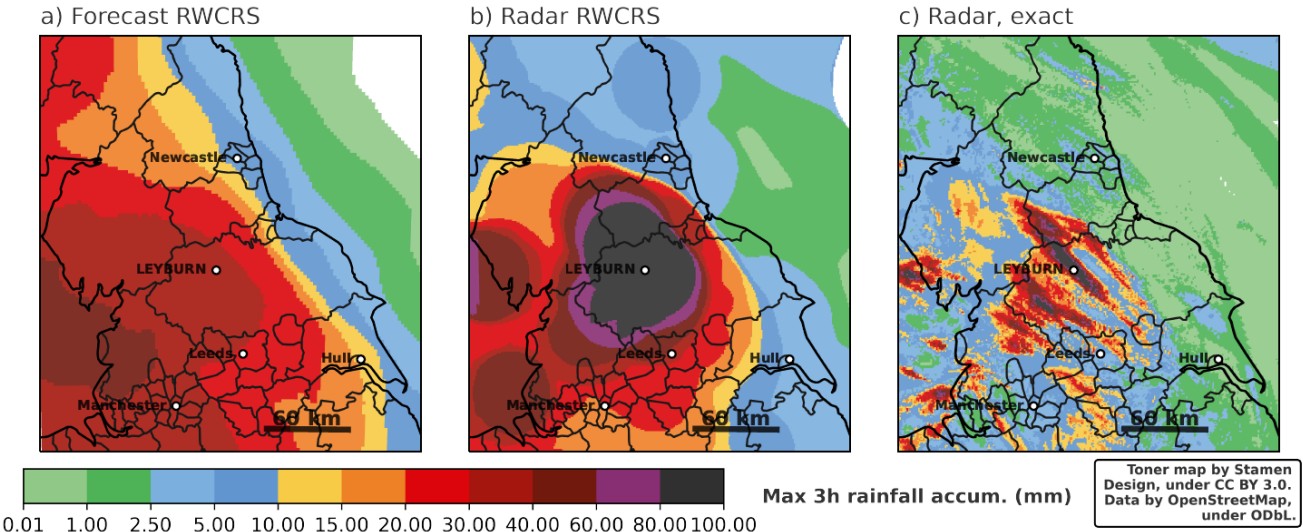

**Figure 1. Forecast and radar observations of maximum daily rainfall accumulation (accum.) in 3 hours for Northern England, valid**
**30/07/2019. (*a*) Accumulations forecast by (*r*30, *p*98, *T*180) RWCRS generated from previous day's 20:00 UTC 18 member**
**MOGREPS-UK ensemble rainfall forecast. (*b*) Benchmark (*r*30, *p*98, *T*180) RWCRS generated from Met Office Nimrod radar**
**observations on 30/07/2019. Labelled town of Leyburn, North Yorkshire, recorded severe SWF impacts. (*c*) Exact (unprocessed)**
**maximum rainfall accumulations in 3 hours, from same radar product.**

### 2.2.2 Flood threshold look-ups

To translate RWCRSs into a SWF forecast we match RWCRS rainfall hyetographs (time-series) with pre-simulated scenarios
by comparing different rainfall thresholds. Specifically, we conduct look-ups of forecast daily RWCRS accumulations against
the Flood Estimation Handbook (FEH) 2022 depth duration frequency (DDF) rainfall curves (Vesuviano, 2022), previous
versions of which underpin the national RoSWF mapping database (Environment Agency, 2019). The FEH rainfall modelling
covers the entire UK at a 10 km grid resolution, both providing the rainfall hyetographs which locally yield a given SWF return
period, and forming the basis for determining return periods for UK rainfall (Stewart et al., 2013). Comparing a forecast
hyetograph for a given location against these values therefore gives an indication of the SWF return period associated with
that rainfall. To identify the flood return period we use a categorical look-up against the DDF threshold values (obtained from
https://fehweb.ceh.ac.uk/Map) for 1 in 5, 10, 30, 100 and 1000 year floods.

The look-up comparison requires sampling a subset of the forecast grid and comparing forecast values against thresholds
indicated by the DDF curves. Advancements in the methods underpinning the FEH rainfall modelling grid have ensured that
recent rainfall extremes are reflected, and that large local variations are minimised and results consistent across the UK
(Vesuviano et al., 2021; Vesuviano, 2022). Considering this and the use of neighbourhood processing used to generate
RWCRSs, local look-up results will remain smooth and can therefore be extrapolated. We adopt local fluvial catchments as a
suitable spatial scale on which to sample rainfall fields. Varied hydrological characteristics means that different catchments
typically respond differently to SWF events, and in the UK are used at an administrative level to define catchment management



activities (Environment Agency, 2009). Here we sample the RWCRS forecasts at the centroid of each Level 9 catchment (roughly 20–70 km catchment scale) from the global HydroBASINS database (Lehner and Grill, 2013), with additional sampling conducted over urban areas. This method provides a hydrologically consistent approach to defining the catchments, sampling and forecast results. Figure S1 shows the locations of the catchment boundaries and sampling points.

At each sample point, a categorical look-up is conducted for daily RWCRS accumulated rainfall in $T=60$, 180 and 360 minutes. The highest associated return period of SWF over all accumulation periods is taken as the day's catchment forecast; similarly, for catchments with multiple sampling points the highest overall return period is selected. In this manner we synthesise the forecasts of multiple RWCRSs, covering multiple time-scales, to provide a single forecast of the SWF risk in each catchment. The output of FOREWARNS is a single guidance map spanning the forecast domain (Fig. 2, column 2) which can be produced

for any MOGREPS-UK ensemble (available hourly). Forecast return periods indicating SWF severity are displayed through categorical colour shading of individual catchments, which are overlaid onto Local Authority boundaries and an Ordnance Survey basemap. Comparison with operational FGS assessments of national flood risk (Fig. 2, column 1) clearly shows the enhanced level of regional detail.

## 2.3 Objective verification

We adopt several methods to verify FOREWARNS forecasts, using a combination of recorded SWF events and radar-derived proxies as our observations (Table 1). We consider two distinct sets of May–October forecasts for Northern England: a sample of days with known, recorded flood events, spanning 2013–2022; and a sample of continuous daily forecasts covering the period of time-lagged MOGREPS-UK forecast availability (2019–2022). To enable continuity across MOGREPS-UK's upgrade from 6 hourly to hourly cycling in 2019, all verification (except Fig. 7) is conducted using 15:00 UTC ensembles.

Under the present system, FOREWARNS forecasts produced from this cycle would be available to users at approximately 19:00 UTC.

### 2.3.1 Flood observations

There is no comprehensive record of UK SWF events, so we construct a record of known May–October days with flood events across Northern England by collating partial SWF records from official reports, news publications and social media. We

primarily use the Global Flood Monitor (GFM; globalfloodmonitor.org), which applies natural language processing to social media activity, identifying all flood types and geolocating likely locations with an accuracy rate of ~90 % (de Bruijn et al., 2019). Given the restricted domain of this study, we manually verified each flood event identified by the GFM, only recording events flagged more than twice that could be subjectively verified as SWF. Due to variability in the available data and the difficulty in interpreting it, we did not quantify the severity or timing of the recorded flooding, but did identify the catchments

where SWF was recorded.

**Figure 2. Forecasts and observations for case studies utilised in the user workshop. Each row pertains to a single case study. The first column shows operational FGS forecasts issued by the Flood Forecasting Centre at 09:30 UTC on the day of the event. Shading indicates flood risk for areas of concern, with yellow indicating "LOW" risk and green indicating "VERY LOW" risk. The second column shows ($r30$, $p98$) FOREWARNS forecasts generated from 20:00 UTC MOGREPS-UK ensemble the previous day. These forecasts would be available to users at 00:00 UTC the same day. Shading indicates expected return period of SWF, from 5 to 1000 years. Areas within the forecast domain with no SWF forecast are bright and unshaded, while areas outside the forecast domain are greyed. The third column shows ($r30$, $p98$) radar SWF proxy observations (shading) and catchment-level locations of recorded floods (bold borders and stippling) for each event. Ordnance Survey MiniScale® basemaps used in columns 2 and 3 contain public sector information licensed under the Open Government Licence v3.0.**



Using both the GFM and official reports, we recorded 82 May–October days with SWF events, spanning 2013–2022 (Table S3). This record represents a lower bound on the actual occurrence of SWF. Events in remote areas or at night may not have
been recorded, while the years of 2013 and 2014 are not covered by the GFM and are based on official reports only.

An upper bound on SWF events may be obtained from a precipitation proxy. This technique is a well-established aspect of US flash flood forecast verification, where proxies are integrated with flood reports in the NOAA Unified Flooding Verification System (UFVS; Erickson *et al.*, 2019; Erickson, Albright and Nelson, 2021). Various such proxies are evaluated in detail in Hermann and Schumacher, 2018. Given the absence of such a UK record, but the availability of national coverage of the
Nimrod radar network and RoSWF mapping, we construct flood proxies for Northern England by using look-ups of radar hyetographs against FEH DDF curves to combine the existing datasets. Since look-ups are conducted at discrete sample points, it is necessary to use neighbourhood-processed radar fields. For consistency we adopt radar RWCRSs as the processing method, and adopt (*r*30, *p*98) FOREWARNS benchmarking driven by radar RWCRSs as radar SWF proxies. See Fig. S3 for verification of multiple parametrisations against catchment-level recorded flood locations. We emphasise that this proxy
should be interpreted as an upper bound on flood occurrence that inevitably overestimates the spatial extent of SWF events. We generate a radar SWF proxy for all 82 days with recorded floods, and daily May–October from 2019 to 2022. Example radar SWF proxies and accompanying catchments with recorded SWF are shown in Fig. 2, column 3.

### 2.3.2 Defining contingency tables

FOREWARNS offers a categorical forecast of the SWF return period for each catchment with its domain. We thus base our
verification methodology on catchment-level yes/no binary contingency tables. We do not attempt to verify exact catchment-level severity, only examining thresholds for all SWF return periods (5 year and above) or severe SWF return periods (30 year and above). A "yes" event occurs in a catchment when SWF severity meeting the threshold is shown by the forecast, event record or radar SWF proxy. If a catchment is highlighted in both forecast and observations, a hit *a* is recorded. Forecast-only "yeses" constitute false alarms *b*; conversely, for observations only we record a miss, *c*. A catchment appearing in neither
forecast nor observations is a correct rejection, *d*.

Assessing forecast performance requires combining *n* contingency tables to obtain a performance record. For Northern England, each FOREWARNS forecast issue predicts the SWF risk for 166 individual catchments. Combining catchment-level contingency tables may be done spatially for each forecast issue, or across all forecast issues for a given catchment. For the former, skill scores pertain to the performance of single forecast issues in predicting the spatial coverage of SWF. We refer to
such measures as *spatial skill scores*. The distribution of such scores across multiple forecast issues then indicates the overall ability of FOREWARNS to accurately warn forecast users of SWF locations. Selecting a single catchment and combining contingency tables across multiple forecast issues meanwhile generates a time-series of forecast performance. We denote measures obtained from such records *temporal skill scores*, since they indicate the relative frequency of forecast "yeses" and misses for a given catchment. The distribution of temporal skill scores across all catchments then indicates the relative
reliability of FOREWARNS.



| Dataset | Source | Details | Application |
|---|---|---|---|
| User survey responses | Debrief survey for workshop participants | • Qualitative responses to questions listed in Table S1.<br>• Survey completed by 20 workshop participants, including 7 forecast users. | • Gathering and evidencing user feedback.<br>• Informing design of verification analysis. |
| Recorded flood locations | • Global Flood Monitor<br>• Official flood reports<br>• News publications | • Northern England flood events identified from May–October, 2013–2022: 82 days with recorded flood.<br>• Catchment-level locations identified.<br>• 28 days with especially significant SWF selected from subjective analysis of recorded flood impacts. | • Identification of days with known flood events.<br>• Selection of radar SWF proxy parametrisation. |
| Radar SWF proxies for days with recorded floods | ($r30$, $p98$) FOREWARNS applied to Met Office Nimrod radar fields. | • Radar SWF proxy generated over Northern England for all 82 days with recorded flood events.<br>• Records categorical SWF return period (None, 5, 10, 30, 100, 1000 year) for 166 catchments. | Objective verification: calculation of spatial skill scores for all 82 days. |
| Daily radar SWF proxy record | Same as row above. | • 725 radar SWF proxies generated over Northern England for May–October, 2019–2022.<br>• 12 days omitted from full period due to radar errors. | Objective verification: calculation of temporal skill scores for all 166 catchments. |
| Subjective, visual forecast assessment | Forecast vs radar SWF proxy pairs | • Human, visual classification of 155 forecast-proxy pairs showing a forecast or proxy "flood". For Northern England, May–October, 2019–2022.<br>• Assessment by 10 meteorologists: 6 authors, 4 external from Met Office and JBA Consulting. | Subjective verification: calculation of integrated skill scores for daily forecast period. |

**Table 1. Summary of datasets and methodologies used in this study for multifaceted verification of FOREWARNS.**

Skill distributions may be aggregated to give single-value measures, for which we report mean values and their standard errors. However, it is important to recognise that such an approach does not give a complete picture of performance: mean temporal skill scores only aggregate point-location assessments, and cannot reflect spatial patterns in forecasts. Likewise, mean spatial skill scores do not provide a clear indication of temporal reliability. For categorical, regional-level warnings such as FOREWARNS, headline score values combining these features may instead be found by reducing each issue to a single, regional contingency table category, generating one time-series of spatially-aggregated performance. There is no unique way to achieve this objectively: one must impose how to define an overall hit, miss or false alarm, and this choice should reflect the requirements of users (Sharpe, 2016). Given the limited observational data available for SWF, we choose to only assess regional-level contingency categories subjectively, from a visual inspection of forecast-proxy pairs.



### 2.3.3 Skill scores

From contingency tables we calculate skill measures covering all aspects of forecast performance (Wilks, 2019). Given the spatial and temporal rarity of flood events, where possible we choose scores which do not reward correct rejections. Such a measure for forecast accuracy is the threat score,

$$TS = \frac{a}{a+b+c},$$ (1)


the proportion of correct forecasts after omitting correct rejections. *TS* is an equitable score, whereby random or constant forecasts are rated equally, typically scoring zero, while perfect forecasts score one. The ability of a forecast to discriminate between different forecast outcomes is meanwhile measured by the hit rate *H* and false alarm rate *F*,

$$H = \frac{a}{a+c}, \quad F = \frac{b}{b+d}.$$ (2)

*H* is equitable and is the probability of detection for a "yes" event. *F* is not equitable (zero for perfect forecasts) and compares the rate of false alarms against the rate of non-events. To instead measure forecast reliability, or the proportion of correct "yes" forecasts, we use the (equitable) success ratio

$$SR = \frac{a}{a+b}.$$ (3)

For a single score measuring all attributes we adopt the Pierce skill score

$$PSS = \frac{ad-bc}{(a+c)(b+d)} = H - F,$$ (4)


for which random forecasts score zero and -1 is the worst score. However, *PSS* can trivially degenerate to *H* for extreme events where the proportion of observed "yes" events, $(a + c)/n$, is very small. This measure has different interpretations for spatial and temporal skill scores. For spatial assessment of single forecasts, the ratio constitutes the spatial coverage *q* for the day in question (i.e. proportion of catchments showing an event).

For temporal records the ratio $(a + c)/n$ instead corresponds to the (climatological) base rate *s*, or the frequency of events for a given location (Stephenson et al., 2008). Figure 3 shows the spatial variation of *s* across Northern England, determined from the daily series of radar SWF proxy observations. SWF does not occur uniformly over the domain, with hillier and wetter western regions experiencing more events (higher *s*). However, values only range from 0.003 (2/725 days) to 0.042 (32/725). A more appropriate skill score in this regime, designed for verifying rare-event forecasts, is the symmetric extremal dependence

index (Ferro and Stephenson, 2011),

$$SEDI = \frac{\log F - \log H - \log(1-F) + \log(1-H)}{\log F + \log H + \log(1-F) + \log(1-H)}.$$ (5)

The behaviour of *SEDI* in the small *s* limit depends on the bias,

$$B = \frac{a+b}{a+c},$$ (6)

or comparison of the average forecast against the average observation. Here a perfect score is one, with larger values denoting

over-forecasting, and values less than one viewed as under-forecasting (Wilks, 2019). *SEDI* is designed to apply to unbiased forecasts. Ferro and Stephenson, 2011, recommend *SEDI* be calculated for forecasts recalibrated such that *B*=1, but



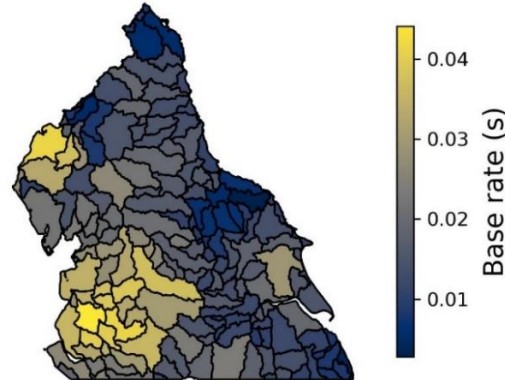

**Figure 3. Spatial distribution of base rate *s* (i.e. proportion of days with events) across Northern England, determined from 725 (*r*30, *p*98) radar SWF proxy observations spanning May–October, 2019–2022. Values range from 0.003 (2/725) to 0.044 (32/725).**

## 3 Forecast co-production

### 3.1 Workshop aims and outcomes

We tested FOREWARNS at a workshop in November 2022 based around three recent, but contrasting, SWF events in Northern England (Table 2 and Fig. 2). There were a total of 21 workshop attendees, who can be split into the broad categories of forecast users, forecast providers, and others. The forecast users included five flood responders and/or flood risk managers from Lead Local Authorities in Northern England, who have primary responsibility for SWF response in their regions; two voluntary Community Flood Wardens; and one Emergency Services representative. The forecast providers included six weather and flood forecasters and/or research staff from the Met Office, Flood Forecasting Centre and Environment Agency. In addition, there were five attendees from UK universities and two from private consultancies.

| Event | Type of region | Date | Severity | Impacts/other details |
|---|---|---|---|---|
| **Case 1:** North-East Yorkshire Dales | Rural area in National Park, with villages and small towns. | 30/07/2019 | Major | Destruction of infrastructure and properties; flooding of properties (including emergency response facilities); widespread flooding of roads. Followed anomalously wet month. Some impacts listed in Kendon, 2019. |
| **Case 2:** Shipley | Small town, suburban river valley. | 30/06/2022 | Minor | Localised flooding of one road and interchange, during evening peak in traffic. |
| **Case 3:** South Yorkshire, North Lincolnshire | Major city (Sheffield), small towns and rural. | 16/08/2022 | Moderate | Flooding of residential roads and properties; damage to roads and property; transport disruption. Followed national heatwave. |

**Table 2. Summary of workshop case studies. The locations of the recorded floods are shown by catchments with bold borders and hatching in column 3 of Figure 2.**





The aims of the workshop were to:

- Gather user feedback on existing SWF forecast provision and understand the actions available to flood risk managers and responders prior to an event.
- Understand how flood responders interpret FOREWARNS, and whether the forecasts would be useful in their decision making.
- Gather suggestions for further improvements to FOREWARNS.

The workshop began with an introduction to FOREWARNS and how it should be interpreted. Participants were then split into four groups for discussions structured by a facilitator. Each case study event was discussed in turn, beginning with some details of the event's impact and discussion of the existing flood forecast provision issued in the four days prior to the event and on the day itself (primarily FGS and NSWWS). The facilitator then introduced FOREWARNS to the group at lead times of four to one day prior to the event, and midnight the night before. A wider discussion between all participants followed, with each group feeding back their main points. At the end of the workshop all participants were asked to fill in a survey (Table S1, question numbers labelled in text).

### 3.1.2 Reflections on existing forecast provision

Operational NSWWS and FGS warnings were valued by workshop participants, and are consulted regularly. Of the seven flood forecast users that completed the survey, four stated that both the NSWWS and FGS products would be either 'useful' or 'very useful' for informing decision making during the three workshop case studies, whilst the remaining three gave the neutral response of 'neither unhelpful nor useful' (Q6 and 7).

The group discussions highlighted that user interpretations of the FGS headline risk statements and maps can differ from interpretations of the written guidance. Experienced flood forecast users placed particular importance on the risk matrices included in these guidance products, far more so than the mapped areas-of-concern, with changes in the risk matrix generating significant attention and often changing actions put in place. The detailed language and descriptions of impacts used in the FGS were also generally considered more useful than the prescriptive "very low" or "low" headline statements, which can sometimes lead to detail within these categories being missed. Indeed, case study 3 was a good example of an event where, although the headline risk level remained constant, the risk matrix indication of likelihood increased with decreasing lead time, which is susceptible to being overlooked.

While getting a national picture of SWF risk is valued, many users reported (Q8) that the broad risk areas, lack of information about the timing of events during a day, and lack of spatial detail make it difficult to apply the information at a local level. The group of forecast users, which mainly consisted of flood professionals, understand the challenges of forecasting convective rainfall and the reasons for false alarms, and have an appetite to access more detailed forecast information with the understanding no forecast is perfect.



### 3.1.3 Actions prior to a flood event

The group discussions of the three case studies highlighted that actions in advance of a forecast flood event are currently limited to a small number of low cost, low regret activities. Two or three days prior to the forecasted event users can monitor forecasts in detail during office hours, clean trash screens to clear urban drainage systems, inform local flood wardens, and alter worker shift patterns to arrange more cover and more on call staff for the event day, particularly if it is over the weekend. One day prior to the event, response vehicles and other equipment can be made ready to deploy, forecasts can be monitored

outside of office hours, and key organisations such as National Highways can be informed. The key period for flood responders is the few hours in the lead up to and during an event, when they monitor radar observations and short lead time forecasts of rainfall via public websites, including the Met Office public webpages. On the same day of an event actions can include notification of the public, closure of public places susceptible to flooding, and more disruptive actions such as road closures.

### 3.1.4 User feedback on the new forecasts

All flood forecast users (7 respondents) reported that FOREWARNS would be 'useful' or 'very useful' for their organisation (Q12), and all users reported that these forecasts were easy to interpret (Q16). Responses indicated forecasts would be used in combination with existing provision on the same day as the event and one day in advance of potential flood events for action planning, including to gain approval from managers for actions, and up to three days in advance for routine monitoring (Q19 and Q20).

For the two more major flood case studies (1 and 3), all flood forecast users stated that they 'agreed' or 'strongly agreed' that the new forecasts would have made a positive difference to their decision making prior to the events (Q9 and Q11). For the minor flood (case study 2, Q10), the responses were spread between 'strongly disagree' and 'agree', with the most frequent response of 'disagree'. The flooding in this case was caused by an extremely intense, but isolated and short-lived, shower, and was minor in extent and impact (i.e. no properties flooded) at a location without a prior history of SWF. From a forecasting

perspective such events were considered impractical to predict for a specific location, while it was clear from discussions that, from a response perspective, such events are not of high concern. Many responders indeed questioned whether the recorded event classified as a flood, and also noted that its impacts could be dealt with reactively. Although participants did not generally find that the FOREWARNS forecast information made a difference to their decision making in this case study, they agreed that NSWWS, FGS, and FOREWARNS all performed well in not flagging any increased risk. A clear outcome from the group

discussions was that the primary concern for flood responders is major SWF events that lead to property damage and widespread disruption. Improving forecasts for such floods should be the priority, rather than trying to anticipate the more minor events.

Flood forecast users were comfortable with the use of a RWCRS and understood that impacts would not occur everywhere marked on the map with a level of elevated SWF risk. There was general agreement that the use of reasonable worst-cases was

better than under-forecasting events. Given participants' understanding of the challenges involved in forecasting SWF events,



there was general tolerance during the group discussions of a certain level of false alarms. False alarms were viewed as less of a concern within the professional community, but more of a concern for the general public (at whom FOREWARNS is not aimed). Users also noted that there were inconsistencies or 'jumpiness' between forecast lead times (e.g. Figure S2, case study 3) and between FOREWARNS and the FGS, which could cause confusion (Q15). Such inconsistences are not isolated to

FOREWARNS (e.g. Speight et al., 2018) and can be alleviated by expert interpretation.

Users felt that the spatial resolution of the forecasts was about right for providing general guidance, given the uncertainties in forecasting SWF events. The users did not see the benefit of the forecasts being at an even higher resolution. Users stated that they would especially value an indication of the time of day of the event, which is not currently provided in the FGS but could be provided as part of the FOREWARNS system. Users said they would routinely check forecasts from about 08 UTC in the

morning, so a forecast issued daily at this time would be most useful, perhaps with an update by lunchtime on days with a forecasted SWF event (note the FOREWARNS flood forecast could be driven by nowcast, rather than NWP, derived RWCRSs at these short lead times).

Several participants stated through the survey (e.g. Q15) and the group discussions that the labelling of flood level by return periods was potentially confusing and could be improved. Users were not necessarily able to relate a certain return period (e.g.

1 in 100 years) to a set of expected impacts, and were unsure whether or not the return periods communicated likelihood or severity. Indeed, the forecasts currently do not explicitly include a measure of likelihood, but rather present a reasonable worst-case scenario. Suggestions during the workshop were to label the different levels simply as high, medium, low severity, and consider what colour scheme is best to represent this.

In summary, aspects of FOREWARNS that were particularly valued by all workshop participants were:

• the improved level of local detail;

   • the clear presentation that is not text heavy and presents the information by river catchment;

   • the translation of rainfall forecasts into a visual flood forecast that indicates impacts, rather than just presenting a weather forecast;

   • use of RWCRSs.

Aspects that participants felt required further improvement were:

   • the communication of flood severity through return periods, since it was not clear what level of impact each return period related to;

   • the appearance of potentially unrealistically high return period values (1 in 1000 year) in some forecasts;

   • the "jumpiness" of forecasts as lead time decreased, and the poorer reliability of the longer lead times;

• the lack of any likelihood indication;

   • provision for, and indication of, the time of day when flooding was expected;

   • the colour scheme – currently potentially misleading.





## 4 Forecast verification



**Figure 4 (cont. next page).**





**Figure 4 (cont.). (*r*30, *p*98) FOREWARNS forecast and radar SWF proxy pairs for 28 days recording especially significant SWF over Northern England, recorded 2013–2022. Locations of recorded floods highlighted in radar proxy panels. All forecasts are based on the previous day's 15:00 UTC MOGREPS-UK ensemble rainfall forecast.**

## 4.1 Verification of forecasts on days with recorded floods

Figure 4 shows pairs of FOREWARNS forecasts and corresponding radar SWF proxies for 28 days where especially significant flood events were recorded. These events are a subset of the 82 days with recorded flooding over Northern England



from May–October, 2013–2022, and represent days where the reported impacts identified damage to property or major disruption. For ($r$30, $p$98) forecasts, SWF was anticipated within the domain for all but 4 events. Radar proxy panels also

display catchments with recorded flooding. The recorded flooding is identified by the proxy in all but two cases (20/07/2014 and 30/09/2017). Catchment-level locations of recorded flood events were forecast in 19 of the events. Subjectively, the forecasts appear to reasonably capture the subdomains at risk of SWF, but overestimate the actual extent compared to the (RWCRS) radar results, which themselves are an upper bound on flood occurrence.

Figure 5 shows spatial skill scores for the forecasts, shown in Fig. 4, of the 28 days where significant SWF was identified.

Scores are plotted on a Roebber performance diagram, where skilful forecasts lie in the top right corner, indicating high forecast accuracy ($TS$) and reliability ($SR$) (Roebber, 2009). Score markers are scaled and shaded based on the SWF coverage $q$, or proportion of catchments highlighted as "yes" in the radar proxy. More than half of the forecasts have $TS > 0.3$ and $H > 0.5$, suggesting good accuracy in forecasting the location of SWF. This is particularly the case for forecasts of events with higher $q$, which generally show higher scores than for lower-coverage events. The slope of a linear regression line with zero intercept

fitted to all points is $B=1.26$, indicating a bias towards over-forecasting SWF extent (versus the radar proxy). The radar proxies for both more major flood events featured in the user workshop are circled (case 1 pink, case 3 yellow).

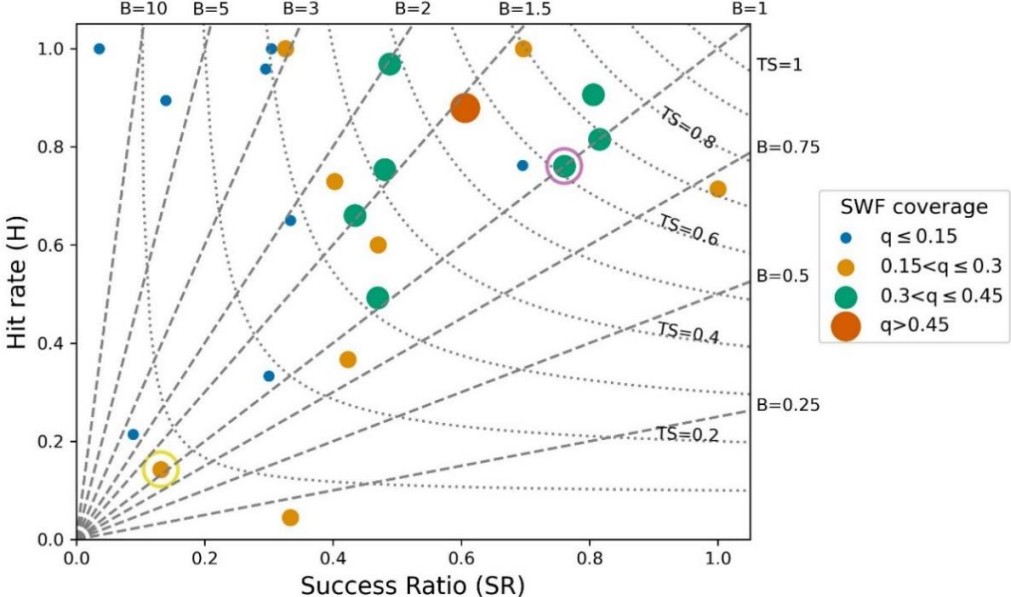

**Figure 5. Roebber performance diagram for ($r$30, $p$98) FOREWARNS forecasts of the 28 days recording significant flood events, 2013–2022. Values plotted are spatial skill scores for individual forecast issues, computed against corresponding radar SWF proxy**

**(as shown in Fig. 4) for all return periods (any indication of SWF). Markers shaded and sized based on proportion $q$ of highlighted catchments in radar proxy. All scores shown are equitable (worst score zero, perfect score one), so forecasts close to the top right corner of the diagram show highest skill. The pink circle indicates the forecast for user workshop case study 1, and the yellow circle case study 3. Case study 2 does not appear because it does not fit the criteria of an especially significant SWF event. All forecasts are based on the 15:00 UTC MOGREPS-UK ensemble forecast issued the day before an event. The 28 days with significant SWF were**

**identified from the full record of 82 days with recorded flooding by subjective analysis of recorded impacts.**



The forecast for workshop case study 1 shows good skill, with *TS*=0.61 and *H*=0.76, whereas scores for case study 3 are far lower (*TS*=0.07 and *H*=0.14). From Fig. 4 the low *TS* values for this case (event 27 in figure) can be seen to arise from an incorrect prediction of the regions at risk of SWF. However, half of recorded flood locations were still successfully identified. Furthermore, the forecasts for case study 3 vary significantly with lead time (see Fig. S2) – at the shortest lead time *TS* increases

to 0.49, with all recorded flood locations identified. The difference in skill between these cases can likely be traced back to differing synoptic-scale regimes: intense rainfall in case 1 was embedded within an intense summer low that tracked across Northern England (Kendon, 2019), whereas in case 3 widespread hydrostatic instability following the breakdown of a national heatwave led to convective precipitation events across much of the UK.

Figure 6 shows distributions of spatial skill scores for *r*30 FOREWARNS forecasts of all 82 days with recorded flooding.

Figure 6a shows results for all return periods, whereas Fig. 6b displays results for return periods higher than 30 years only, which we refer to as severe SWF (note that this is at the catchment level). Both figures show the importance of using a high percentile threshold: for *p*90 the interquartile range for most scores is nearly zero, and the same applies for *p*95 for severe SWF. However, the higher percentiles show far larger spreads for the false alarm scores (*F* and *SR*), especially *p*99. While this percentile achieves higher hit rates than *p98* for both return period thresholds, the increased cost of spatial false alarms is

reflected in the far less pronounced improvement in *TS* and *PSS* between these percentiles. The scores calculated here are taken against the radar SWF proxy, so false alarm measures likely represent lower bounds on truth values. With these considerations, the improvement in hit rate from *p*98 to *p*99 is arguably not sufficient to justify the increased occurrence of spatial false alarms. Note that the effect on forecast performance of changing *p* is far more pronounced than changing neighbourhood radius *r*, with

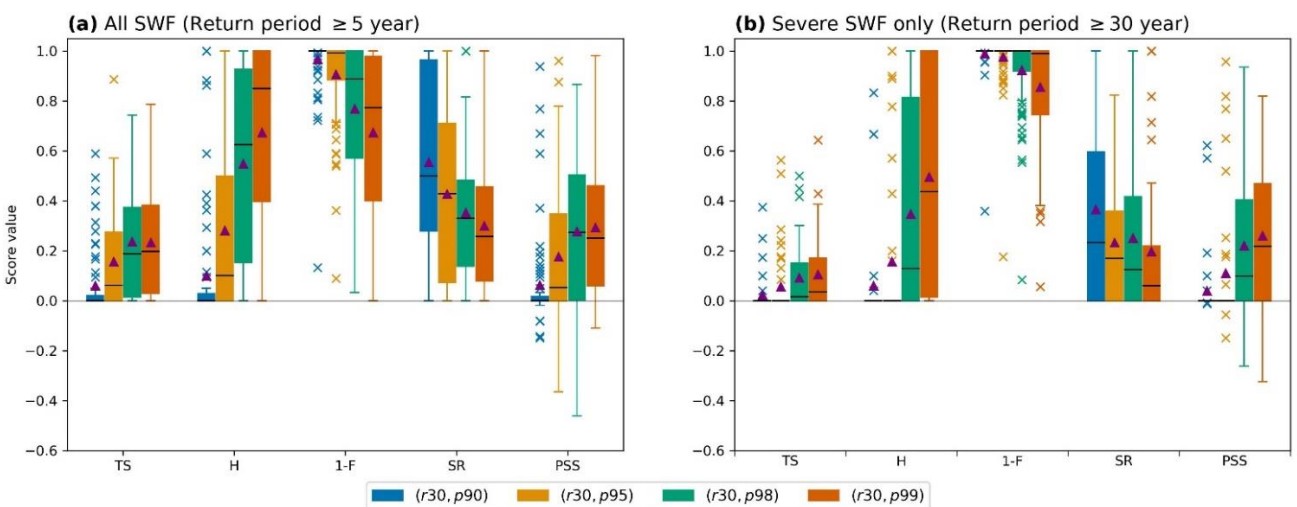

**Figure 6. Distributions of spatial skill scores for *r*30 FOREWARNS forecasts of all 82 days with recorded floods, grouped by increasing percentile. (*a*) Distributions for *TS*, *H*, 1-*F*, *SR* and *PSS*, calculated against (*r*30, *p*98) radar SWF proxy for all return periods. Shading indicates percentile, while mean (median) values are shown by purple triangles (black horizontal lines). All measures are equitable (worst score zero, perfect score one) except *PSS*, for which random forecasts score zero and worst score is -1. Each distribution calculated from 82 forecasts based on 15:00 UTC MOGREPS-UK ensemble issued day before a recorded flood**

**event in Northern England between May–October, 2013–2022. (*b*) Repeated for severe SWF return periods (30 year or higher) only.**


Fig. S4 showing that changing *r* has little effect on mean skill, compared to changing *p*. Based on this analysis, we focus the remainder of our evaluation on the (*r*30, *p*98) FOREWARNS parametrisation.

Figure 7 shows how spatial skill distributions for (*r*30, *p*98) forecasts change for a range of lead times when calculated for the 41 days since 2019 with recorded flood events. Here we use a smaller sample size since prior to 2019, MOGREPS-UK

produced ensembles every 6 hours, with maximum lead time of (up to) only 54 hours. Post 2019, the most up-to-date MOGREPS-UK ensemble available for issuing a complete FOREWARNS forecast (i.e. full calendar day from 00:00 UTC) is the previous 20:00 UTC cycle. This 5 hour decrease in lead time (blue bars) is sufficient to slightly improve both the distribution spread and average values for the hit rate (*H*) and *PSS*. Conversely, there are generally slight decreases for forecasts available at longer lead times. As may be anticipated, forecasts available 4 days in advance of an event (purple bars) show the

poorest skill, especially in *H* and *PSS*, but overall the results show remarkably consistent spatial forecast skill with lead time.

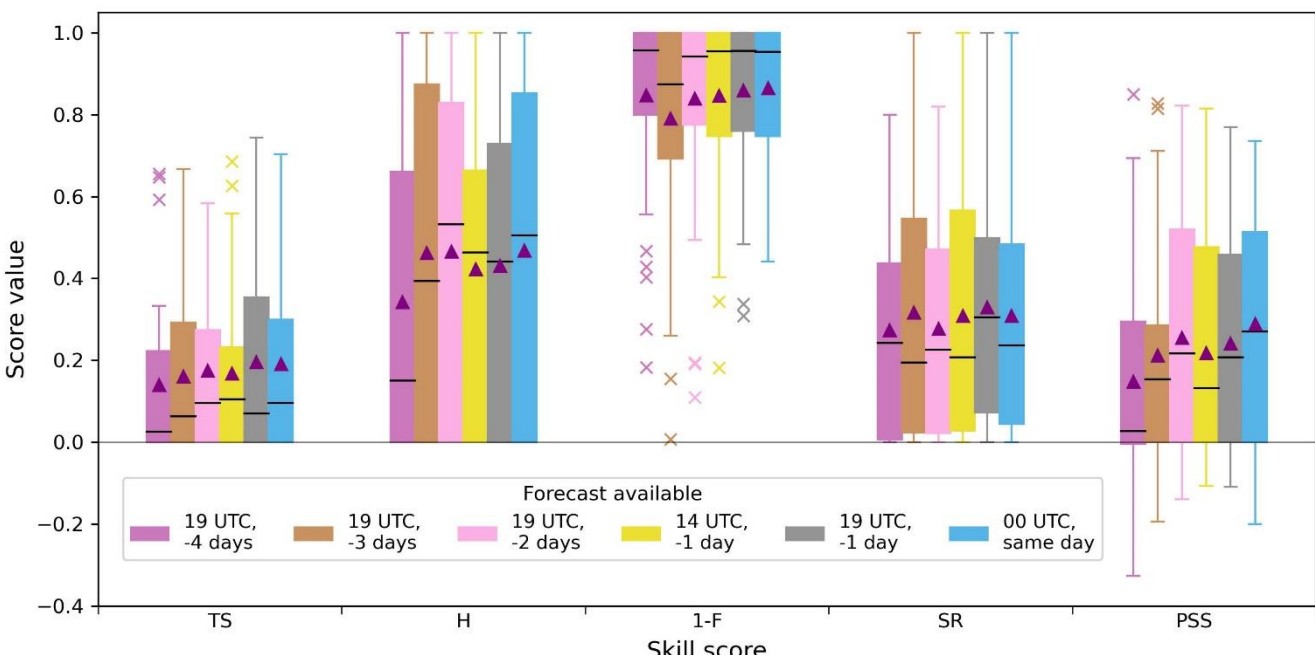

**Figure 7. Distributions of spatial *TS*, *H*, 1-*F*, *SR* and *PSS* for (*r*30, *p*98) FOREWARNS forecasts of all 41 days with recorded flooding since 2019, grouped by decreasing lead time. Scores calculated against (*r*30, *p*98) radar SWF proxy for all return periods. Shading indicates time FOREWARNS would be available to users and lead time in days prior to a recorded flood event in Northern England**
**between May–October, 2019–2022. Forecasts available at 19:00, 14:00, 00:00 UTC are based on MOGREPS-UK ensemble forecasts from 15:00, 10:00 and 20:00 UTC respectively. Distributions calculated from 41 forecasts, with mean (median) values shown by purple triangles (black horizontal lines). All measures are equitable (worst score zero, perfect score one) except *PSS*, for which random forecasts score 0 and the worst score is -1.**

### 4.3 Verification of daily forecasts

The previous results assess the spatial skill of FOREWARNS for 82 days with recorded flood events. Here we assess 725 daily forecasts and radar SWF proxies (May–October, 2019–2022, 12 days omitted due to radar errors). There are 41 days during





this period with recorded SWF events, of which 12 were missed by the radar proxy. An additional 79 days showed SWF in the radar proxy, yielding 108 proxy flood days in total. Across the same period, (*r*30, *p*98) FOREWARNS forecast SWF in at least one catchment on 107 days. There was at least one catchment-level hit against the radar proxy on 52 of these days, versus 570

days which exclusively showed correct rejections. The remaining 103 days showed only catchment-level false alarms and/or misses. Figure S5 shows that across all days there was a bias towards over-forecasting the spatial extent of SWF.

Figure 8 shows plots of objective temporal hit rates *H* against *F* (ROC diagram) and *SR* (Roebber diagram), i.e. one point for each of the 166 individual catchments in the forecast domain. Corresponding distributions, and results for spatial skill scores from this forecast sample, are shown in Fig. S6. Figure 8a shows that for all catchments *F* is extremely low, whereas *H* spans

a wide range of values, with mean value 0.37±0.01. In only six catchments do we find that SWF was not correctly forecast at all (*H*=0). The *F* values show a low rate of erroneous warnings for days with no flooding – the frequency of forecast issues reflects the temporal sparsity of flood events, indicating very good discrimination. However, *F* depends on the correct rejection rate and does not reflect the reliability of event forecasts.

The distribution of *SR* values plotted in Fig. 8b indicates that overall, the probability a catchment-level "yes" forecast of SWF

is correct is relatively low (mean 0.26±0.01). However, when we account for the variability in the base rate *s* (i.e. proportion of days with events) of SWF across Northern England (Fig. 3), we see relatively consistent *H* values, but a notable improvement

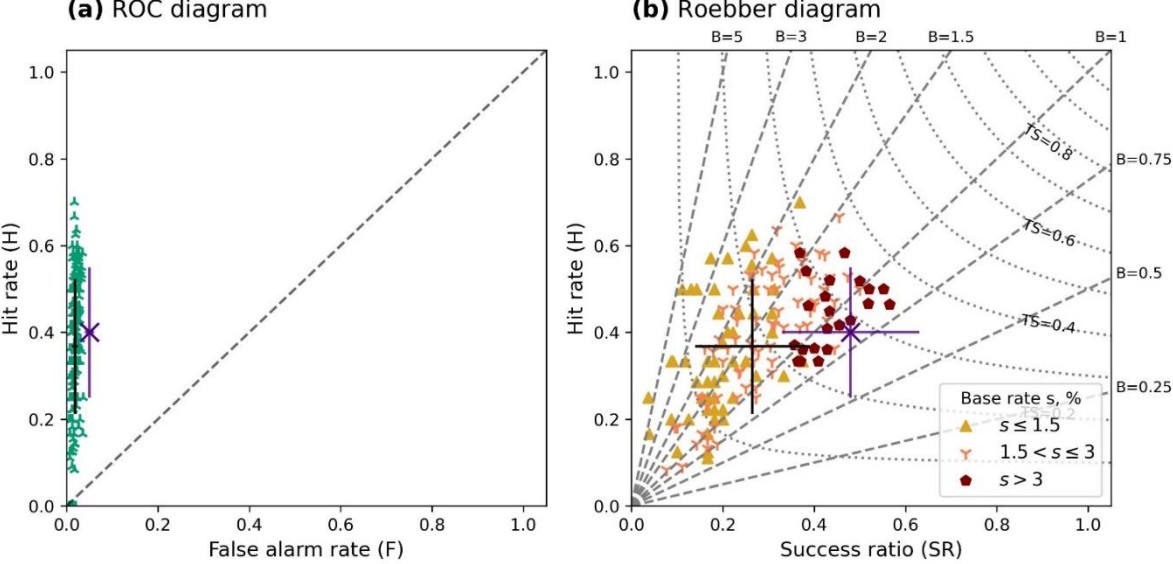

**Figure 8. Temporal skill scores for (*r*30, *p*98) FOREWARNS forecasts. (*a*) ROC diagram for scores from 166 catchments in Northern England, calculated for 725 daily forecast issues against radar SWF proxy, May-October 2019 – 2022. All forecasts are based on**

**15:00 UTC MOGREPS-UK ensemble, where FOREWARNS would be available to users at approximately 19:00 UTC, and valid next day. Perfect forecasts lie towards the top left of the diagram. Mean forecast (*F, H*) values plotted (black), with error bars indicating standard deviation. Mean values from subjective verification indicated by purple cross (error bars also show standard deviation). Standard deviations for *F* are included but extremely low. (*b*) Roebber diagram for same forecast sample; perfect forecasts lie towards top right of diagram. Shading and marker style indicate base rate *s* for each catchment. Mean forecast (*SR, H*)**

**values plotted (black), with error bars indicating standard deviation. Mean (*SR, H*) values from subjective verification indicated by purple cross (error bars also show standard deviation).**



in *SR* for catchments which have higher base rates. Indeed, for catchments with $s>3\%$ the mean *SR* increases to $0.44\pm0.01$, while mean $H=0.44\pm0.02$. This indicates that forecasts are generally more accurate for catchments which experience more SWF events. The improvement in accuracy also corresponds to a general decrease in the degree of over-forecasting, as

indicated by a trend towards the unit bias line. Figure S7 shows that the spatial variability of many temporal skill scores also strongly reflects that of *s* (Fig. 3).

The plots in Fig. 8 comprise objective results for individual catchments assessed in isolation, and do not account for the overall spatial distribution of any given day's forecast. To include spatial skill in the assessment of the daily forecast series we use results from a subjective assessment (see Sect. 2.3.2). A group of 10 meteorologists (six paper authors, four practitioners from

Met Office and JBA Consulting) visually examined all forecast-proxy pairs from May–October, 2019–2022 that did not exclusively show correct rejections (i.e. 155 pairs). Individual assessments and subsequent skill scores are given in Table S2, which shows a mean miss count of $50.2\pm3.2$, compared to mean false alarm count of $34.7\pm1.7$, suggesting regional misses were prioritised over false alarms. This implies that from a human perspective, accounting for both forecasts' spatial distribution and temporal frequency, SWF is viewed as under-forecast over Northern England. Indeed, the mean subjective

bias $B=0.81\pm0.04$ is less than one.

Mean subjective skill scores include hit rate $H=0.40\pm0.05$ and reliability $SR=0.48\pm0.05$ (purple, Fig. 8b). These scores are not directly comparable to their objective counterparts, since subjective scores reflect the spatial distribution of forecasts, whereas mean objective scores are derived from individual catchments. However, from spatial skill assessment in Sect. 4.1 we know that forecasts overestimate the extent of SWF, and over an extended period this will lead to temporal false alarms. The

improved subjective scores (versus objective results, Fig. 8b) reflect the value humans placed on forecast spatial patterns of SWF over individual catchment-level comparisons.

Figure 9 shows the distributions of individual catchments' objective temporal skill scores at a range of lead times. Grey bars correspond to the one-day lead time used for the preceding objective and subjective assessments (Fig. 8). For all scores we see a decrease in skill with longer lead time, which is more marked than that shown in Fig. 7 for the spatial skill of recorded flood

events (2013–2022). Catchment-level accuracy measures, as indicated by *TS*, are generally quite low, with maximum score 0.37 at one-day lead time, reflecting the tendency towards spatial over-forecasting. However, at this lead time *PSS* shows good levels of skill in nearly all catchments, reflecting the very low *F* values in Fig. 8a. Since *PSS* scores random forecasts zero, we also note that at all lead times FOREWARNS performs better than a random SWF forecast nearly everywhere across Northern England – at one-day lead time the only exceptions are the six catchments where $H=0$ (identified in Fig. S7f). Changes of

spatial skill distributions for these daily forecasts are shown in Fig. S8, and as with Fig. 7, show lower variability in skill than the temporal scores shown here. This suggests consistent predictions for the occurrence of convective rainfall by the input MOGREPS-UK ensemble, but with decreasing uncertainty in location as lead time decreases.

The very low base rates *s* of SWF observed across Northern England require that skill scores be interpreted with caution, as many measures take trivial values in the low *s* limit (Stephenson et al., 2008). We therefore include distributions for the *SEDI*

score, which does not degenerate in this regime (Ferro and Stephenson, 2011). The distributions of *SEDI* values plotted in Fig.



9 are universally positive, thereby again showing performance better than random at all lead times. At one-day lead time *SEDI* has a high mean value of 0.61±0.01, and maximum 0.85. Since the *SEDI* score is designed to apply to unbiased forecasts (*B*~1), at all lead times we have excluded catchments with anomalously high bias values from the distributions shown here. Although difficult to translate into an intuitive interpretation of forecast performance, the *SEDI* score values clearly reinforce

our finding that, despite the rarity of SWF events, FOREWARNS provides a skilful forecast of the catchment-level hazard.

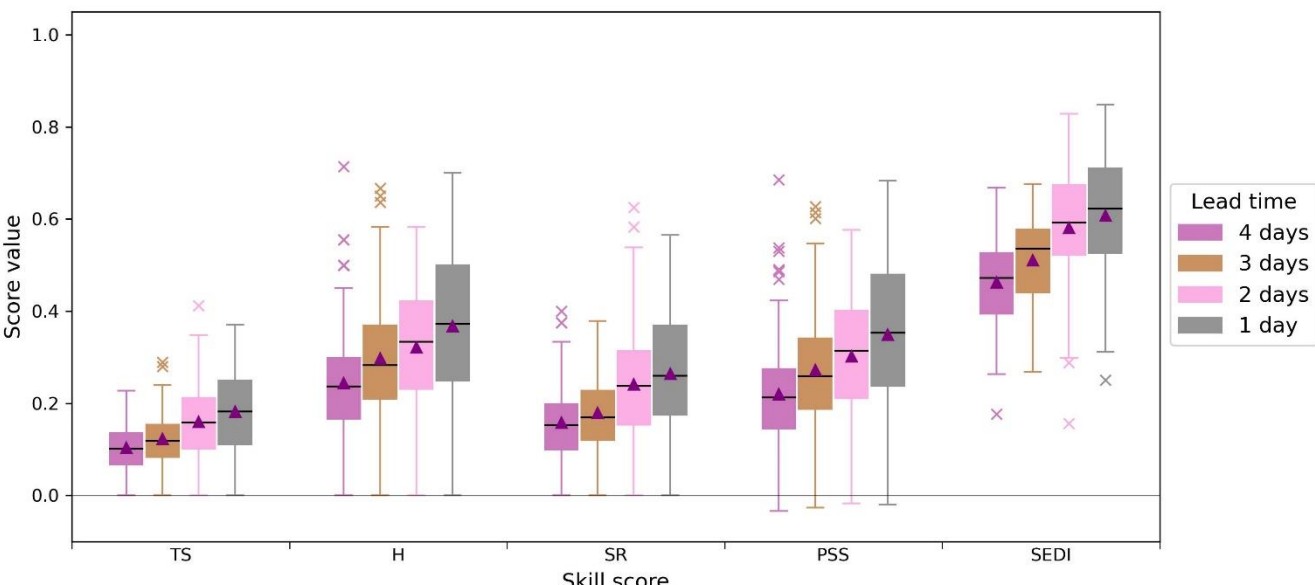

**Figure 9. Distributions of temporal *TS*, *H*, *SR, PSS* and SEDI for (*r*30, *p*98) FOREWARNS forecasts available to users at approximately 19:00 UTC (based on 15:00 UTC MOGREPS-UK ensemble) and grouped by decreasing lead time. Scores computed for all return periods against radar SWF proxy on forecast validity date for 725 days, May–October 2019–2022. Shading indicates**
**lead time in days, with scores for forecasts issued at one day lead time shaded grey. Distributions calculated from forecasts for 166 catchments, with mean (median) values shown by purple triangles (black horizontal lines). All measures are equitable (worst score zero, perfect score one) except *PSS* and *SEDI*, for which random forecasts score zero and the worst score is -1. Distributions for *SEDI* exclude catchments where *B*<0.67 or *B*>1.5.**

## 5 Summary and recommendations

The UK is underprepared to deal with convective rainfall events (Greater London Authority, 2022). As the climate changes, such events are predicted to intensify and occur more frequently in many parts of the UK; improving tools and capabilities to increase preparedness is essential. In this paper we have demonstrated and tested a novel SWF forecasting method, FOREWARNS (*Flood f**ORE**casts for surface **WA**ter at a **R**egio**N**al **S**cale*). FOREWARNS combines neighbourhood-processed ensemble NWP rainfall forecasts with pre-simulated hydrological modelling to provide SWF forecasts for a

reasonable worst-case rainfall scenario (RWCRS). We have examined whether FOREWARNS meets the requirements of UK forecast users through a workshop structured around facilitated group discussions of three case study events in Northern England. We have objectively assessed FOREWARNS' performance by conducting verification over this region, examining aspects of spatial and temporal performance for recorded flood days and continuous proxy records.





Although FOREWARNS aims to meet a UK user-need for enhanced regional-level SWF forecasts that complement existing
national guidance (Ochoa-Rodríguez et al., 2018; Birch et al., 2021), the methodology and findings are not geographically
restricted. Indeed, FOREWARNS would complement existing efforts to improve global flash flood guidance (Georgakakos et
al., 2022). RWCRSs may be derived from any convection-permitting forecast system (Böing et al., 2020), while the threshold
look-ups used to derive flood impacts require only a database of flood records or pre-simulated scenarios. Here we have used
FEH DDF modelling underpinning UK RoSWF maps (Vesuviano, 2022), but datasets such as NOAA's gauge-data derived
Atlas-14 product for the contiguous United States could be adopted (described in Herman and Schumacher, 2018). Meanwhile,
the verification of any SWF forecast faces the same issues of poor observational datasets and temporal sparsity faced here.
Our solution, a combination of recorded events and radar proxy observations of SWF, extends previous verification of US
flash flood guidance (Erickson et al., 2019) and could be applied anywhere with high-resolution radar observations, or globally
through satellite-derived rainfall estimates. For the UK a priority should be to build on this approach and create an integrated
dataset combining SWF records and radar/gauge proxies, similar to the NOAA Unified Flooding Verification System
(Erickson et al., 2021).

We conducted objective verification of FOREWARNS against radar SWF proxy observations for 82 May–October days with
recorded SWF events from 2013–2022, and a 725 day continuous period spanning May–October, 2019–2022. From this
verification we recommend adopting the ($r30$, $p98$) RWCRS parametrisation. FOREWARNS demonstrates high spatial hit
rates (mean $H$=0.55±0.04) for days with recorded flood events, demonstrating that just over half of proxy flood locations were
forecast. Since the radar SWF proxy overestimates flood extent, the real hit rate is likely higher as there will be fewer misses.
This skill is, in part, inherited from the MOGREPS-UK system (Porson et al., 2020) and will change if other rainfall ensembles
are applied to the forecasting system instead. The catchment-level severity of SWF is typically underestimated, with lower hit
rates for return periods of 30 years or higher. Overall spatial reliability scores are considerably lower (mean $SR$=0.35±0.03),
with nearly two in three catchment-level "yes" forecasts being false alarms, but this is to be expected given the application of
RWCRSs. The workshop found that forecast users understood and supported the use of reasonable worst-case forecasts,
prioritising the successful prediction of SWF over minimisation of false alarm rates. Actions considered by users in response
to forecasts are typically low cost, but require a lead time of a day or greater. Given the potential impacts of SWF, false alarms
thus carry far lower costs for communities than forecast misses.

The frequency of occurrence of SWF events is low across the region but does show spatial variation, with more events in
hillier and wetter western areas. The temporal false alarm rate of FOREWARNS for any given location is extremely small and
reflects the relative frequency of events in Northern English catchments. The temporal hit rate of forecasts for single locations
does not vary significantly by catchment, but FOREWARNS tends to forecast SWF too often in catchments where (proxy)
flooding occurs less frequently. FOREWARNS forecasts show excellent values for the *SEDI* score designed to evaluate
forecasts of extreme, temporally rare events, with values for all catchments (and at all lead-times) higher than for a random
forecast. Under a visual, subjective verification combining spatial and temporal performance, a tendency for FOREWARNS
to miss regional-scale SWF events was recorded. This reinforces that correctly predicting the regional hazard necessitates



issuing false alarms for any given point-location; what is important is that this rate remains at a similar order of magnitude to the low frequency of events, which is the case here.

In a workshop survey all flood forecast users stated FOREWARNS would be 'useful' or 'very useful' for their organisation. Respondents foresaw using the forecasts in combination with existing national provision for action planning on the same day as and one day in advance of potential flood events, especially where major flooding is forecast. The subsequent quantitative verification demonstrates that user confidence in FOREWARNS is warranted. Although few verification studies of comparable forecasts are published, and none are directly comparable, FOREWARNS' performance compares favourably with the

literature. Roebber diagrams for spatial skill of extreme rainfall forecasts analysed in Erikson et al., 2019 show notably lower skill than apparent here, while Erikson et al., 2021 report comparable mean temporal skill scores (specifically $H$ and $TS$) for NOAA Extreme Rainfall Outlook forecasts, but with higher false alarm rates. Mean $SEDI$ scores are slightly lower than values previously reported for extreme rainfall forecasts (North et al., 2013; Sharpe et al., 2018), reflecting the additional complexities of forecasting SWF.

Specific results support the adoption of FOREWARNS for users' specific requirements: the strongest forecast skill is evident at short lead times (1 day ahead), and although the spatial skill of individual forecasts can vary markedly, in general skill is higher for more significant events with higher spatial coverage. For operational purposes the FOREWARNS system verified here does not issue forecasts at suitable times, but it would be possible to issue forecasts at any hour for the following day. Based on the workshop discussions, forecasts issued roughly midday the day before, and updated early the next morning,

would better satisfy the needs of users. An ideal system would then issue rainfall nowcast-derived warnings (instead of using ensemble NWP rainfall forecasts) where appropriate, fulfilling the user need for very short lead time forecasts.

One issue not addressed in this study is forecasting probabilities of predicted outcomes. Although FOREWARNS is ostensibly a categorical forecast of SWF risk, the underlying ensemble RWCRSs are inherently probabilistic. However, the forecasts do not communicate an overall probability of a given RWCRS occurring. Given the high uncertainties in forecasting SWF, a

priority for development should be the accurate calculation of this information, and its dissemination with forecasts. The time within the 24 hour forecast period that the SWF events may occur is additional valuable information that can be provided from the RWCRSs. Forecast users stated in the workshop discussions that time-of-day information would be extremely valuable, and verifying this additional feature should be the subject of future work.

We have shown both qualitatively and quantitatively that FOREWARNS has the potential to meet the needs of UK forecast

users as an operational system, if integrated with future national SWF forecast provision in the manner illustrated in Fig. 10.

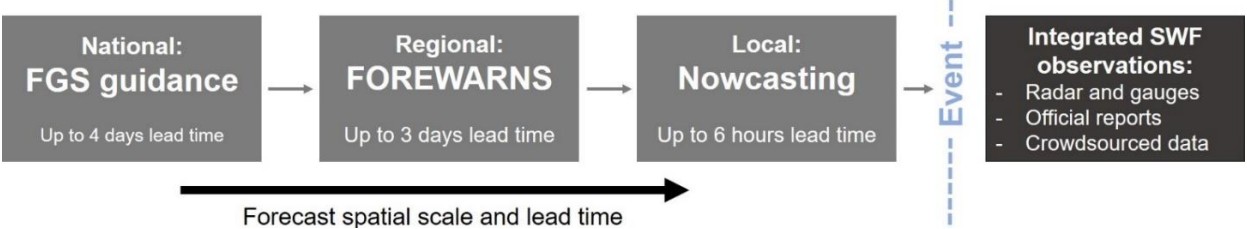

**Figure 10: Summary of envisioned place of FOREWARNS within future UK SWF guidance available to flood responders.**



In summary, our specific recommendations for the further development, operationalisation and dissemination of FOREWARNS include:

- Using features of the existing RWCRS method, modify FOREWARNS to communicate the time of day an event will occur, and with what likelihood. These features should be evaluated.

- Review the use of return periods (1 in 5, 10, 30, 100 and 1000 years) as the indicator of catchment-level SWF severity within FOREWARNS. Consider the use of a colour scale labelled by "low, moderate, high, very high severity" instead.

- Consider how to ensure that FOREWARNS is consistent with the national guidance issued within the FGS, particularly around the 'jumpiness' of some FOREWARNS issues at different lead times.

- FOREWARNS is intended for experienced, trained forecast users only and should be disseminated either by a password protected platform, or via email alongside other national provision such as the Flood Guidance Statement.

- Forecasts should be issued daily in the morning for the following one to three days, and could include more regular
updates for days where a SWF event is predicted.

- Further research is required to assess whether or not it is appropriate to issue FOREWARNS updates with a lead time of 6 hours or less that are driven by rainfall nowcasts, or if real-time inundation modelling (e.g. Birch et al., 2021) is more appropriate at these timescales.

- Expand the region covered by FOREWARNS to all of England and Wales, and test in a pseudo-operational setting
alongside operational SWFHIM and FGS guidance over a summer season (planned for 2023).

- Alongside the development of FOREWARNS we have illustrated the use of quantitative, qualitative and combined verification methods that address the challenges of limited data. To enable improved SWF forecast verification we recommend developing a complete, consistent historical record of the timing, extent and severity of SWF events over England and Wales.

**Acknowledgements**

This study was conducted as part of the Yorkshire Integrated Catchment Solutions Programme (iCASP, NERC grant NE/P011160/1). We are extremely grateful to all participants of the November 2022 workshop for their invaluable contributions to this study, and to Jennifer Bonner, Emma Cowan and Melanie Stonard for their support in organising this. We are also grateful to Christine McKenna and Heather Forbes of JBA Consulting, and David Hayter and Brent Walker of the Met
Office, for their assistance in conducting subjective forecast verification. We have benefited enormously from wider collaborative discussions with the Met Office and Flood Forecasting Centre, and additionally thank Emily Barker, Katherine Egan, Bill Leathes, Rachel North, Julia Perez, David Price, Nigel Roberts and Adrian Semple in particular. We acknowledge use of the Global Flood Monitor (globalfloodmonitor.org) for compiling records of known flood events; HydroBASINS database (hydrosheds.org/products/hydrobasins) for defining fluvial catchments; and the FEH Web Service





(fehweb.ceh.ac.uk/Map) for obtaining rainfall modelling data, using code written by Richard Rigby of CEMAC, University of

Leeds. This work also used JASMIN, the UK collaborative data analysis facility, and the Met Office MASS data archive.

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
