# Peer review of "FOREWARNS: Development and multifaceted verification of enhanced regional-scale surface water flood forecasts"

_Natural Hazards and Earth System Sciences, 2023_

## Author Response (AR1)

**Referee 1**

General comment

Thank you for your comments and review. We are pleased that you view our work as being of valuable interest to the wider community, and have endeavoured to improve our description of how initial soil moistures are taken into account. You are entirely correct that there is no assimilation of soil moisture state; the FEH modelling data used as a hydrological reference is a reference data product generated assuming idealised storm profiles. We state that this data is generated from modelling (Vesuviano, 2022) – however, we have amended the text to make it explicit that no current soil moisture observations are used by FOREWARNS, and highlight this in the discussion.

We fully acknowledge the suggestion that switching the ordering of Sections 3 and 4 may be beneficial – indeed, this was a debate among the authors when writing the draft. However, we have respectfully maintained the original ordering: we feel it is essential that the verification only be described in full in the context of the needs and requirements of intended users. The results of the verification analysis on their own do not give a good indication of the forecast performance, given the lack of benchmark literature to compare against and the limited availability of observations. This aspect of the study was conducted with the needs of users in mind, and we ultimately chose to reflect this in the section ordering. Note also that Section 3 stands entirely independent of Section 4, making no reference to results therein; this is not true in reverse.

Detailed comments

L39-40: *« severity and frequency » What is the distinction here ? In the text after, you seem to make no distinction. Please homogeneise the vocabulary (see also rq . L141).*

As noted, severity is used in this paper to denote event frequency. To avoid confusion the word "severity" here has been replaced by "impacts".

L110: *« Tennant, 2015 » : this reference is missing*

Thank you for catching this, the appropriate reference has been added.

L141: *« severity » : Say clearly that for you, severity = frequency, expressed in term of return period. Note that this definition does not take into account the "risk"but only the "hazard" (see also rq L39-40, where you seems to do the distinction.)*

As suggested, we have amended this paragraph to add a final sentence clarifying our meaning of "severity". The sentence now reads "Note that by severe events we mean temporal extremes associated with high rainfall return periods – we do not assess hazard impact or potential damages."

L147: *« multiple ensemble member fields » : Please explain how do you deal with ensemble to obtain one unique FOREWARNS forecast ? In all the study, it seems you deal with deterministic forecasts (see fig 1 or 2 for example).*

To obtain a unique, apparently deterministic forecast field from an ensemble, percentile sampling is conducted across multiple ensemble rainfall fields, as noted in the text, building a single rainfall distribution representative of the ensemble forecast. To make this clearer we have amended the text to read: "The processing may be conducted either on a single rainfall field, or across multiple ensemble member fields (covering common forecast periods) by sampling the distribution of maximum accumulations generated by all ensemble members – see Böing et al, 2020. Any RCWRS is then parametrised as *(r, p, T)*. The timings…"

L165: *Please indicate the legend legend for black contours. Local authority boundaries ?*

Thank you for highlighting this; the black contours are indeed local authority boundaries. The legend has been amended to note this.

L170-178: *this part is very unclear. You don't explain how exactly your flood return periods are obtained. You explain how you pass from a multi-frequency hyetogram to a mono frequency (using your national scenario), but you don't explain how you pass from rainfall to flood (since rainfall return period is not equal to flood return period, it also depends on initial condition on the catchement). If I understand, you don't use G2G runoff outputs (runoff) as in SWFHIM… is it correct ?*

To clarify, as we are interested in surface water flooding, we are using the rainfall return period rather than the flood return period, which is typically estimated from discharge levels. This is consistent with surface water hazard mapping approaches, such as the RoFSW datasets produced by the Environment Agency. We have added a brief explanation at the end of this paragraph (i.e. first of S2.2.2) stating:

"It is important to note that by taking this approach, we are only considering the return period associated with the rainfall, rather than estimating the return period associated with river flow or discharge, which will be impacted by multiple processes, including antecedent conditions. This is consistent with approaches used to mapping surface water flooding, such as the UK Risk of Flooding from Surface Water Maps (Environment Agency, 2019)"

L186: *why do you choose the centroid ? Wouldn't be possible to study all the return period estimated using your method, and then take the higest return period in each catchment ?*

The choice of cell catchment was made firstly based on evaluating multiple points within the catchment to determine the relative sensitivity, and secondly to ensure consistency across all catchments, which vary in size and orientation, and with other surface water mapping approaches. We found that the choice of sampling locations within a catchment was relatively insignificant, due to the smooth underlying profile of the FEH rainfall descriptors. As is mentioned at Line 180, this dataset has been designed to be smooth across the UK so as to ensure that large variations do not occur, which would have an impact on the application and quality of the FEH datasets. The use of cell centroids in surface water modelling is consistent with the development of the UK RoFSW maps, which involved determining the rainfall at the model domain centroid, and applying this to the 10km$^2$ model domain area.

To provide the reader with some explanation of this we have made the following amendment at the end of this section: "This method provides a hydrologically consistent approach to defining the catchments, sampling and forecast results, whilst the use of centroid locations to determine return periods is consistent with the development of other surface water datasets, such as the RoFSW maps (Environment Agency, 2019). Figure S1 shows…"

L187-188: *What is the physical motivation in the definition of those catchments. In particular, why do you choose smaller catchments only in urban areas (and not for all) ? Why do you say it is « hydrologically consistent » ? What exactly is behind this agregation / sampling procedure in term of final results (just for mapping, or more deep consequencecies on the results?)*

We emphasise that smaller catchments are not used over urban areas; rather, we take additional sampling locations to reflect the greater exposed populations within these catchments. The catchment boundaries used are always the Level 9 Hydrobasins dataset.

By hydrologically consistent we mean that this is a catchment dataset constructed at a common hierarchical level using a consistent, well documented and accepted methodology, and moreover that representation of the hazard is consistent within the physical boundary that it would occur in: the catchment watershed. We have chosen this particular level of the Hydrobasins dataset as the best fit to the scale of the RWCRS rainfall fields being used as an input rainfall field (i.e. ~60km).

In terms of final results, flood forecasts are displayed on the catchment level, with the issued forecast depending on the maximum sampled rainfall within each catchment. By providing extra sampling over urban areas, but not using smaller catchments, we therefore avoid undue bias in our results towards these areas, but do account for the practical forecasting need to capture any raised forecast risk over urban areas.

L200: *« MOGREPS-UK ensemble » : could you explain how do you deal with different members to obtain one unique forecast (see also rq L147)*

See reply to comment on L147; the amendments provide greater clarity to the reader on how the ensemble is reduced to a single field.

L213: *« more than twice that could be subjectively verified as SWF » : What do you mean by that ? What is your "subjective" definition of SWF ? A return period > 5 year ?*

By subjective verification here we mean a purely qualitative examination of FGM data entries, which had to be led by expert interpretation. The FGM, as a resource scraping social media, is itself entirely qualitative and only provides images and textual descriptions of flood events, which are themselves generated by the public or media. It was therefore not possible to assign any quantitative flood return period to identified events. Our assessment of the FGM was instead intended to identify events (of which no sufficiently comprehensive record exists in the UK for our study period) that could be used for verification analysis.

We have changed the wording here to avoid reference to "verification", and thus avoid confusion with results in Section 4. The text has been amended to: "that could be subjectively identified as SWF based on expert judgement".

L220: *Figure 2 : the quality of the image should be improved (difficult to read the FGS column)*

We apologise for the lower quality in this image. The final figure itself is a high resolution pdf image, however we were not able to embed this within the initial manuscript submission. Please be assured that the final figure exists, and is higher resolution.

L235: *« RoSWF » : I don t understand how RoSWF have been used in this study. For me it has not been used but I maybe misunderstood (same for G2G, see my remark L170-178).*

The RoFSW mapping is an official database generated by using model FEH DDF rainfall curves. FOREWARNS makes use of the same underlying model data; this is made explicit in Section 2.2.2 (which has been amended to improve clarity). We have chosen to refer to RoFSW here as more readers will be familiar with the national coverage provided by the headline RoFSW product.

L238: *« Fig S3 » : this figure is cited before S2*

The offset numbering is deliberate; Fig. S2 pertains to specific information provided to users in the User Workshop. Fig. S3 meanwhile relates to specifics of the verification choices.

L240: *« inevitably overestimates » : For me, there is also 2 explanations for this strong over estimation (median at 90% of catchments in "false alarm" if I understand correctly Fig S3) :*

We agree with your comments regarding the overestimation inherent within our proxy SWF observations method, and have amend the text to emphasise the absence of soil moisture observations. The limitations of this observational substitute are noted throughout the text, and intended as a proxy only. We note that in Fig S3 the results for the false alarm rate are plotted as 1 - $F$, such that a median of 90% corresponds to a good, i.e. **low**, false alarm rate.

The text has been amended to read: "that inevitably overestimates the spatial extend of SWF events. The proxy measure does not account for antecedent hydrological conditions or any intensity of flood damage, and should not be considered a replacement for realistic, but expensive, hydraulic modelling. We generate…"

L251: *« contingency tables » : please say that n = nb of day for temporal, and nb of catchment for spatial verification (if I correctly undersand)*

To address this point we have modified a slightly later sentence. The third sentence of the paragraph now reads "Combining catchment-level contingency tables may be done spatially for each forecast issue, such that $n$ is the number of catchments, or across all forecast issues for a given catchment, such that $n$ is the number of forecasted days."

L270: *« from a visual inspection of forecast-proxy pairs » : This is the only explanation you give. It is not possible for the reader to understand what has been made (see remark L524 ).*

We would note that the preceding sentences in this paragraph outline the problem obstructing any analysis of combined spatial and temporal skill, contextualising this description of our chosen method. However, to aid clarity we have added the following sentence to make our methodology more explicit:

"we choose to only asses regional-level contingency categories subjectively. Forecast issues are characterised by using the individual expert judgement of multiple meteorologists to assign a unique category based on a visual inspection of forecast-proxy pairs."

L276-282: *« equitable » : I understand that the number of d (true negatives) are not accounted, but please explain it is more « equitable ».*

Here equitable is meant as a definition: a forecast score is equitable if random or constant forecasts are rated equally, typically scoring zero, while forecasts score one (Wilks, 2019). The text has been amended to make this clear.

L305: *figure 3 : Please indicate your threshold defining an event (5 or 30-year)?*

The threshold here is 5 years. The first parentheses in the caption have been modified to read "i.e. proportion of days with events at a 5 year return period or higher".

L307-412: *this part is very interesting. However, I would put it after the results of 4., and I would try to make it shorter. Indeed, there is already an interesting discussion about the usefulness of the method (from an user point of view) that should be said at the end, rather than here.*

See General Comments.

L424: *«for all but 4 events. R » : Which ones ? Which criteria ? Is it the same visual analysis than later (see remark L524)*

The 4 days are those with no forecast catchments: 29/07/2013, 30/09/2017, 16/05/2021 and 30/09/2021. This is solely based on examination of Figure 4, and is separate to the later visual analysis. We do not feel it is necessary or useful to write these dates explicitly in the text. However, to avoid confusion with the later analysis the sentence has been modified to "From the figure it is apparent that for *(r30, p98)* forecasts SWF was anticipated within the domain for all but 4 events."

L437: *Figure 5 : Please indicate your threshold defining an event (5 or 30-year)?*

The 5 year threshold is implied by the statement "for all return periods"; we now explicitly state that this is 5 years or higher.

*Do we to have 28 points of the figure (one by day) ? Because I see only 24.*

There are only 24 data points as *H* and *SR* are trivially zero for a forecast in which no hits were forecast. The legend has been amended to state that trivial forecasts are omitted.

L448: *«half of recorded flood locations were still successfully » : Do you see this result on figure 4, event 27 for the forecast (left column) ? I see only one recorded flood location correctly identified. I maybe misunderstood.*

You are absolutely correct, this statement is incorrect and was based on a forecast lead time that was not shown in the final draft. This sentence has been deleted, thank you.

L489: *« 4.3 » : should be 4.2 ?*

Correct, this has been amended; thank you.

L491-493: *There are … in total » : it is easier to understand if you change this by : « There are 41 days during this period with recorded SWF events, of which \*\*29 were correctly detected\*\* by the radar proxy. An additional 79 days showed SWF in the radar proxy, yielding 108 proxy flood days in total. »*

*Bis: this means that for 79 over the 108 « proxy flood days » no flood has been « observed/reported ». If I correctly understood, this seems huge. For me, this is because you don't take into account initial soil moisture conditions AND damage impacts in your method (see my rq L240)*

Advised textual amendment has been made. Regarding the 79 of 108 proxy flood days on which no flood has been observed/reported, we agree that the value is large and indicative of issues underlying the proxy method. This is why we are careful to refer to the proxy observations as an upper bound on flood occurrence. We do expect a significant number of floods to go unreported due to their occurrence in unpopulated areas or lack of damage, but quantifying that expectation is impossible without greater research into SWF observations (a need we highlight in our Discussion).

L498: *« for spatial skill scores » : replace by « for spatial \*\* and temporal \*\* skill scores*

Amendment has been made.

L522-523: *« do not account for the overall spatial distribution » : if I understood you gave spatial scores into Fig 6S, first row.*

You are correct that spatial scores are given in Fig. S6. However the results in Fig 8 are independent, having being obtained from a temporal contingency table record for each catchment in isolation. The statement that the figure does not account for the overall spatial distribution of a day's forecast is correct and meant to emphasise this. This also provides the

context for the need for a visual, subjective assessment of combined spatial and temporal skill.

L524: *« see Sect. 2.3.2 » : there is very few explanation in section 2.3.2. (see remark L270). I don t understand the results. Could you please give more details after line 270 ?*

See reply to remark L270.

L565 – end: *« Summary and Recommendations » should be more nuanced (taking into account my remarks L240, L399-412, L491-493 bis...)*

We thank you for your comments and have made multiple amendments, following your recommendations, to add nuance to the discussion:

"require only a database of flood records or pre-simulated scenarios. FOREWARNS does not account for antecedent hydrological conditions and is intended to complement forecast systems based on hydraulic models, limiting its application but also minimising hydrological data input, which may often be unavailable. Here we have used…"

"Our solution, a combination of recorded events and radar proxy observations of SWF providing lower and upper bounds on flood hazards from severe rainfall, extends previous verification of US flash flood guidance.

…NOAA Unified Flooding Verification System (Erickson et al., 2021). Our methods do not attempt to verify the impact of flood damage, which should be also be included in such a resource.

**Referee 2**

General comment

Thank you for your comments and review, we are glad that the results are interesting, especially regarding our workshop with users. We accept the remarks regarding the clarity of the Methods, particularly Section 2.2.2, and have made amendments to this section following your suggestions. Thank you for bringing this to our attention. We further hope that links will be included within the document in a finalised version, as we agree that this would aid navigation.

Regarding the suggestion to switch the ordering of Sections 3 and 4 – this was a debate among the authors when writing the draft. However, we will respectfully maintain the current ordering: we feel it is essential that the verification only be described in full in the context of the needs and requirements of intended users. The results of the verification analysis on their own do not give a good indication of the forecast performance, given the lack of benchmark literature to compare against and the limited availably of observations. This aspect of the study was conducted with the needs of users in mind, and we ultimately chose to reflect this in the section ordering. Note also that Section 3 stands entirely independent of Section 4, making no reference to results therein; this is not true in reverse.

Specific comments

L104: *What do you mean by significant impacts? What is their nature? Which return period?*

Significant impacts here are mentioned in the specific context of another warning system, the National Severe Weather Warning Service. These impacts hold for severe weather hazards, as stated in the text, and do not have a fixed return period. We do not consider the precise definition used by NSWWS relevant to this study evaluating an independent forecast product, but to avoid any confusion have changed the word "significant", which is ascribed a definition later on, to "substantial".

L110: *The reference Tennant, 2015 is missing in the bibliography*

Thank you for catching this, the appropriate reference will be added.

L138-141: *As for lines 104-105, what do you mean by severity?*

Severity here means event frequency. To make this clear an additional sentence has been added to the end of the paragraph, reading "Note that by severe events we mean temporal extremes associated with high return periods – we do not assess hazard impact or potential damages." The word "severity" has also been changed to "impacts" in L39 to maintain consistency with this definition.

L173-174: *Is a rainfall grid with a resolution of 10 km sufficient for the local/regional scale? In general, this paragraph is not very clear regarding the choice of precipitation thresholds for different return periods.*

As we have referenced in other parts of the paper (Line 180), the FEH Rainfall statistics are designed to be smooth across the UK to remove large variations between local areas. In the presentation of the FEH methods, the use of 10km is determined to be sufficient to remove local variations, whilst being consistent with the evaluation of rainfall return periods across the UK. This approach ensures that a consistency across the UK as a whole.

L188-189: *Why is this method considered consistent? It lacks justification and/or references. Why use the value corresponding to the watershed centroid? How is this more relevant than the maximum return period over the entire watershed?*

We consider the use of catchments as being consistent with the realisation of an extreme rainfall hazard in a region of land. Typically, surface water will impact a catchment, rather than a broader area (for example, Boscastle in 2004). Other approaches to modelling surface water hazards include using regular grid cells as the model domain. The issue with this approach is that cells can cover multiple catchments, with the degree to which rainfall affects each cell in turn varying considerably. By considering catchments rather than grids, we are providing a more consistent approach to the hydrology of the hazard, which is what we are referencing here.

The other point raised here is the use of cell centroids, rather than multiple sampling points. As we have mentioned in our response to a similar point raised by Reviewer 1, the use of cell centroids is based on being consistent with previous surface water mapping projects (RoFSW in the UK), and across the catchments which vary in size and orientation. When testing the method we found that the choice of sampling locations within a catchment was relatively insignificant. Furthermore, the FEH dataset from which we sample rainfall return periods is quite smooth across each catchment, with little internal variation, so the impact of this approach is low.

We have referenced this with the following amendment to this paragraph: "This method provides a hydrologically consistent approach to defining the catchments, sampling and forecast results, whilst the use of centroid locations to determine return periods is consistent

with the development of other surface water datasets, such as the RoFSW maps (Environment Agency, 2019). Figure S1 shows…"

L200: *How is the transition from a probabilistic forecast (ensemble forecast lines 147-148) to a deterministic forecast achieved?*

To obtain a unique, apparently deterministic forecast field from an ensemble percentile sampling is conducted across multiple ensemble rainfall fields, as noted earlier in the text, building a single rainfall distribution representative of the ensemble forecast. To make this clearer we have amended the text at L147-148 to read: "The processing may be conducted either on a single rainfall field, or across multiple ensemble member fields (covering common forecast periods) by sampling the distribution of maximum accumulations generated by all ensemble members – see Böing et al, 2020. Any RCWRS is then parametrised as *(r, p, T).* The timings…"

L213-213: *How is a flooding event differentiated from a SWF ? What are the criteria?*

The identification of events here was carried out from a purely qualitative examination of FGM data entries, which had to be led by expert interpretation. The FGM, as a resource scraping social media, is itself entirely qualitative and only provides images and textual descriptions of flood events, which are themselves generated by the public or media. By investigating all entries on the FGM for a given event, and seeking other external resources, we used our judgement to differentiate SWF and fluvial flood events; for example, excluding cases that were clearly on flood plains and connected to an overtopped fluvial channel. In some cases floods did have a fluvial element, but there were also clear independent pluvial events involved. Of course, such a discrimination can never be fully precise. Our assessment of the FGM was merely intended to identify events (of which no sufficiently comprehensive record exists in the UK for our study period) that could be used for verification analysis, and does not itself represent a verification step.

We have changed the wording here to avoid reference to "verification", and thus avoid confusion with results in Section 4. The text has been amended to "that could be subjectively identified as SWF based on expert judgement".

L239-241: *It is interesting to note the overestimation; however, it lacks explanation. Is it due to methodological choices, especially the choice of a certain rainfall threshold? Does the initial state of the watershed not influence the response to precipitation? The absence of consideration for the initial moisture of the watershed could also explain this overestimation. What about the quality of event selection, and can we be sure that the GFM method does not underestimate the number of SWF events?*

We have amended the text to comment further on the overestimation inherent in the proxy, adding the sentence "The proxy measure does not account for antecedent hydrological conditions or any intensity of flood damage, and should not be considered a replacement for realistic, but expensive, hydraulic modelling."

As noted in the text already, the GFM method is by necessity a lower bound on flood occurrence – it will absolutely underestimate the number of SWF events. The proxy, conversely, will then overestimate the number of events, with true flood occurrence lying somewhere within those bounds. Our methods acknowledge this inherent problem caused by current observational constraints, which we later argue must be improved.

L251: *This sentence would benefit from being clearer by defining "n" beforehand (number of forecast days / number of watersheds).*

We have amended a later sentence (third sentence of paragraph) to make the definition of *n* clearer. This reads "Combining catchment-level contingency tables may be done spatially for each forecast issue, such that *n* is the number of catchments, or across all forecast issues for a given catchment, such that *n* is the number of forecasted days."

L276: *I am not really sure to understand what is the meaning of equitable*

Here equitable is meant as a definition: a forecast score is equitable if random or constant forecasts are rated equally, typically scoring zero, while forecasts score one (Wilks, 2019). The text will be amended to make this clear.

L426: *How is the "good" localization of events qualified? Visually and due to the overestimation, Figure 4 suggests that there are fewer than 19 cases of good localization.*

The figure of 19 forecasts reflects the number of cases in which there is any forecast of a recorded flood location. Later sentences then caveat that this will be through some overestimation, as requested. We also note that the recorded flood locations will often represent an underestimation; again there is the balance between the lower and upper bounds on occurrence.

*Section 5 – This section lacks a bit of critical reflection on the methodological choices made.*

We have made multiple amendments to add further nuance and critical reflection to the discussion:

"require only a database of flood records or pre-simulated scenarios. FOREWARNS does not account for antecedent hydrological conditions and is intended to complement forecast systems based on hydraulic models, limiting its application but also minimising hydrological data input, which may often be unavailable. Here we have used…"

"Our solution, a combination of recorded events and radar proxy observations of SWF providing lower and upper bounds on flood hazards from severe rainfall, extends previous verification of US flash flood guidance.

…NOAA Unified Flooding Verification System (Erickson et al., 2021). Our methods do not attempt to verify the impact of flood damage, which should be also be included in such a resource.

Remarks on figures

Fig 1: *Lack of legend, what is the black line ? District boundaries ?*

The black lines are Local Authority boundaries. The legend has been amended to describe this.

Fig 2: *Reading the content of the first column is relatively difficult.*

We apologise for the lower quality in this image. The final figure itself is a high resolution pdf image, however we were not able to embed this within the initial manuscript submission. Please be assured that the final figure exists, and is higher resolution.

Fig 3: *For which threshold ? Severe SWF or just SWF ?*

The threshold here is 5 years, i.e. all SWF. The first parentheses in the caption will be modified to read "i.e. proportion of days with events at a 5 year return period or higher".

Fig 5: *Mistake concerning the label ? Only 24 points on Fig 5 and 28 days in the label…*

There are only 24 data points as *H* and *SR* are trivially zero for a forecast in which no hits were forecast. The legend will be amended to state that trivial forecasts are omitted.